# *Smooth Reading*: Bridging the Gap of Recurrent LLM to Self-Attention LLM on Long-Context Understanding

**Kai Liu**
Tongji University* & Shanghai AI Laboratory
liukai1998@tongji.edu.cn

**Zhan Su**
Halmstad University
zhan.su@hh.se

**Peijie Dong**
Hong Kong University of Science and Technology (GZ)
pdong212@connect.hkust-gz.edu.cn

**Fengran Mo**
University of Montreal
fengran.mo@umontreal.ca

**Jianfei Gao & Shaoting Zhang & Kai Chen**
Shanghai AI Laboratory
{gaojianfei, zhangshaoting, chenkai}@pjlab.org.cn

## Abstract

Recurrent large language models (Recurrent LLMs) offer linear computational complexity as efficient alternatives to quadratic self-attention-based LLMs (Self-Attention LLMs). However, Recurrent LLMs underperform on long-context tasks due to limited fixed-size memory. Previous research focused on architectural innovations to enhance memory capacity, but failed to match Self-Attention LLM performance. We argue this limitation stems from processing entire contexts at once being ill-suited for Recurrent LLMs. We propose *Smooth Reading*, a co-design of recurrent architecture and inference method. It introduces a end-to-end multi-round inference method that processes context incrementally and iteratively summarizes information, reducing memory demands. Methodologically, we reveal architecture-inference interactions play an important role for performance, efficiency and scalability, shedding light on future Recurrent LLM design. Besides, our method substantially bridges the performance gap between Recurrent and Self-Attention LLMs on long-context tasks while preserving efficiency advantages. *Smooth Reading* boosts SWA-3B-4k from 5.68% lower to 3.61% higher performance than Self-Attention LLMs on LongBench, while maintaining 2.5× faster training and 2× faster inference at 64k context.

## 1 Introduction

Self-Attention-based large language models (Self-Attention LLMs) have demonstrated strong long-context understanding capabilities (Bai et al., 2024; Hsieh et al., 2024). However, as applications increasingly require longer contexts—such as complex reasoning (DeepSeek-AI & et al., 2025), embodied agents (Zhang et al., 2024a), and research (Zheng et al., 2025)—the quadratic computational cost of self-attention becomes a major barrier to scaling. In response, Recurrent LLMs have re-emerged as a promising alternative (Peng et al.; Gu & Dao; Yang et al., 2025; 2024; Sun et al.; Yang et al.), offering linear-time computation and constant memory.

We categorize mature LLM use cases by context length into: (1) general chat (short context), (2) long chain-of-thought (CoT) reasoning (long output), and (3) long-context understanding (long input). Recurrent LLMs match Self-Attention LLMs on short-context tasks (Yang et al., 2025; 2024; Yang et al.; Dao & Gu) and long CoT reasoning (Zhao et al., 2025), but lag substantially behind on

---

*Kai Liu is with the Shanghai Research Institute for Intelligent Autonomous Systems, Tongji University.

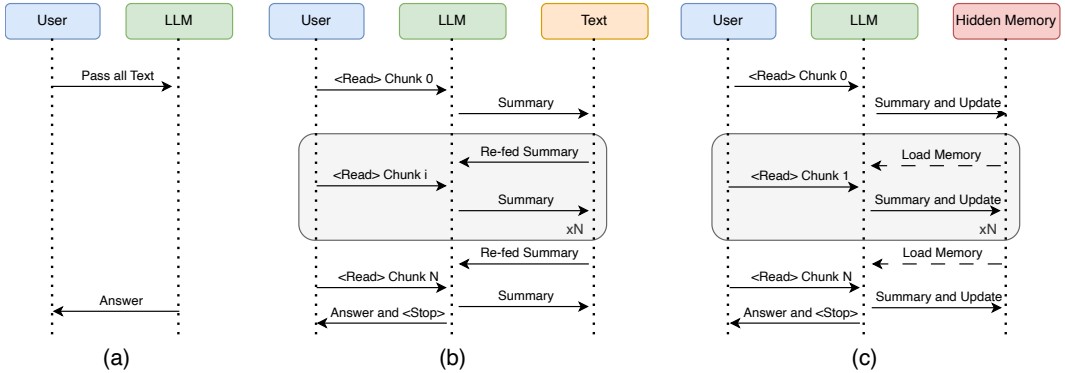

Figure 2: Comparison of inference methods. (a) One-Round inference (OR): process the entire context in a single pass; (b) Non-End-to-End Multi-Round inference (NMR): chunked processing with summaries re-inserted into the prompt; (c) End-to-End Multi-Round inference (EMR): iterative chunk reading with hidden memory updates preserved across rounds.

long-context understanding (Yang et al., 2024; Waleffe et al., 2024; Liu et al., 2025). **This gap is a primary obstacle to the wider adoption of Recurrent LLMs.**

Most prior work attempts to close this gap through architectural changes that increase memory capacity (Peng et al., 2024; Yang et al., 2025; Peng et al., 2025; Dao & Gu; Qin et al.; Sun et al.; Liu et al., 2025), such as more expressive update rules (Yang et al., 2025) or larger state sizes (Liu et al., 2025). Despite progress, these approaches alone have not matched the performance of Self-Attention LLMs. A key reason is the misalignment between architecture and inference: traditional One-Round (OR) inference (Figure 2a)—processing the entire context in a single pass—demands much larger memory than Recurrent LLMs can provide.

We address this limitation with *Smooth Reading*, a co-design method to optimize both architecture and inference method jointly for Recurrent LLMs. Our approach consists of three main components:
(1) A tailored EMR procedure for Recurrent LLMs that processes long inputs chunk-by-chunk. After reading each chunk, the model produces a short contextual summary and updates its hidden state, preserving and refining hidden memory across rounds. This compresses information into a local working window and avoids overloading fixed-size memory. Thanks to the linear efficiency of Recurrent LLMs, the Multi-Round overhead remains manageable and still more efficient than Self-Attention LLMs. (2) An architectural perspective showing that different inference strategies favor different properties: while OR prefers maximal single-pass memory capacity, EMR benefits more from strong length extrapolation (length generalization). It demonstrates the importance of inference method in architecture design. (3) A joint analysis of architecture and inference that yields practical guidelines for balancing performance and efficiency, enabling optimal co-design.

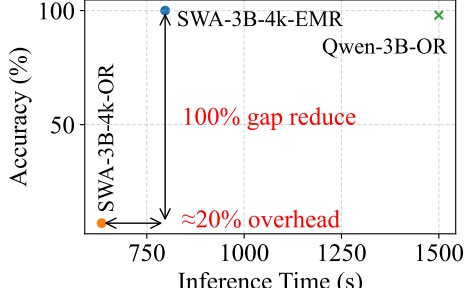

Figure 1: Inference Time vs. Accuracy on NIAH at 64k. Applying EMR to Recurrent LLM (SWA-3B-4k) significantly closes the gap to Self-Attention LLMs (Qwen-2.5-3B-OR) while incurring only a 20% overhead over OR inference, and remains about 2× faster than Self-Attention LLMs.

We evaluate architecture-inference combinations on LongBench (Bai et al., 2024), Needle-in-a-Haystack (Hsieh et al., 2024), RULER (Hsieh et al., 2024), and HELMET (Yen et al., 2025). Smooth Reading provides a favorable accuracy-efficiency trade-off. As shown in Figure 1, Smooth Reading significantly narrows the performance gap between Recurrent LLMs (SWA-3B-4k-EMR) and Self-Attention LLMs (Qwen-2.5-3B-OR) on long-context tasks while incurring only a 20% overhead over One-Round inference (SWA-3b-4k-OR) and remains about 2× faster than Self-Attention

counterparts. On real-world long-context tasks, Smooth Reading boosts SWA-3B-4k from 5.68% lower to 3.61% higher performance than Self-Attention LLMs on LongBench. These results indicate the significance of architecture-inference co-design for Recurrent LLMs.

In summary, our contributions are:

- While Recurrent LLMs suffer on long-context understanding tasks, our Smooth Reading method substantially closes the performance gap with Self-Attention LLMs while preserving efficiency advantages. It significantly pushes the boundary of Recurrent LLMs and makes them more practical for real-world applications.

- We demonstrate the importance of architecture-inference co-design, showing how their interaction governs performance, efficiency, and scalability, and provide actionable guidelines beyond architecture-only improvements.

## 2 RELATED WORK

Most prior work improves either model architectures or inference strategies in isolation. Inference is typically designed around Self-Attention LLMs, whereas new Recurrent LLMs are often evaluated under traditional One-Round inference, leading to a mismatch between architecture and inference.

### 2.1 SELF-ATTENTION LLMS AND INFERENCE STRATEGIES

**Self-Attention LLMs.** Self-attention (Vaswani, 2017) underpins most modern LLMs (DeepSeek-AI & et al., 2025; Qwen et al., 2025; Achiam et al., 2023). By attending over all preceding tokens, these models can, in principle, use an arbitrarily long context. However, the cost scales quadratically in computation and linearly in space with input length, making very long contexts inefficient.

**One-Round inference.** The standard approach for Self-Attention LLMs is a single forward pass over the entire input (Figure 2a). This aligns with their ability to attend to all prior tokens, but incurs $O(L^2)$ computation and $O(L)$ space, where $L$ is the input length, limiting scalability.

**End-to-End Multi-Round inference.** End-to-End Multi-Round inference methods split the input into chunks and iteratively process them, often appending intermediate summaries back into the prompt (Figure 2c). This increases the effective context to $\lambda L$ tokens ($\lambda > 1$), yielding $O((\lambda L)^2)$ computation and $O(\lambda L)$ space. For Self-Attention LLMs, this compounds the already high complexity.

**Non-End-to-End Multi-Round inference.** Methods such as prompt compression or chain-of-thought prompting (Zhang et al., 2024b; Yoon et al., 2024; Qian et al., 2024; Lee et al., 2024; Chen et al., 2023) discard hidden states and re-feed a compressed context at each step to cap the per-step length (Figure 2b). This yields $O(\lambda L)$ computation and $O(1)$ memory. However, discarding hidden states causes information loss and can degrade performance.

Table 1 summarizes these strategies for Self-Attention LLMs. Despite its quadratic cost, One-Round inference remains prevalent due to its simplicity and strong performance. End-to-End Multi-Round inference is rarely used with Self-Attention LLMs on long-context tasks due to its high complexity.

### 2.2 RECURRENT LLMS

Recurrent LLMs use fixed-size memory, enabling linear-time and constant-space inference regardless of input length. This efficiency arises from architectural choices such as linear-attention variants or sliding-window mechanisms (Beltagy et al., 2020). However, most recurrent models are still used with standard One-Round inference, which can overwhelm their fixed memory on long-context tasks and limit performance. Recent efforts target these limitations by (1) increasing memory capacity (Qin et al.; Gu & Dao; Dao & Gu; Du et al., 2025) or (2) improving memory efficiency (Yang et al., 2025; Sun et al.; Yang et al., 2024). While helpful, Recurrent LLMs still trail Self-Attention LLMs on tasks requiring long-context understanding.

Table 1: Comparison of architecture-inference combinations. "C" and "S" denote computational and space complexity. "$L$" is the input length. "$\lambda$" ($\lambda > 1$) is a scaling factor for the effective context length in Multi-Round settings. "OM" indicates whether the inference strategy overwhelms the model's memory. "HM" indicates whether hidden memory is preserved during inference. Both memory overwhelm and discarding hidden memory degrade performance. The main drawback of each combination is underlined. Our Smooth Reading (End-to-End Multi-Round inference with Recurrent LLMs) achieves the best balance between performance and efficiency.

| Architecture | One-Round | Non-End-to-End Multi-Round | End-to-End Multi-Round |
|---|---|---|---|
| Self-Attention | C: $O(L^2)$; S: $O(L)$ 
 OM: No; HM: — | C: $O(\lambda L)$; S: $O(1)$ 
 OM: No; HM: No | C: $O((\lambda L)^2)$; S: $O(\lambda L)$ 
 OM: No; HM: Yes |
| Recurrent | C: $O(L)$; S: $O(1)$ 
 OM: Yes; HM: — | C: $O(\lambda L)$; S: $O(1)$ 
 OM: No; HM: No | **C: $O(\lambda L)$; S: $O(1)$** 
 **OM: No; HM: Yes** |

## 2.3 Architecture-Inference Co-Design

The interaction between architecture and inference is seldom examined, yet it is crucial for Recurrent LLMs with fixed memory. We analyze this interplay and propose an inference method tailored to Recurrent LLMs, leveraging their strengths while addressing their constraints. We demonstrate the interaction of architecture and inference significantly influences performance, efficiency, and scalability. A concurrent line of work, OPRM (Ben-Kish et al., 2025), adapts retrieval-augmented generation to mitigate the memory overflow problem in linear-attention architectures. This highlights the importance of inference design for Recurrent LLMs. However, OPRM does not systematically study architecture-inference co-design, which can limit performance and generalization.

## 3 Methodology

The interaction between model architecture and inference strategy is critical for long-context performance. We first review the characteristics of Recurrent LLMs, then introduce *Smooth Reading*, which comprises:

1. An End-to-End Multi-Round (EMR) inference procedure tailored to Recurrent LLMs;
2. Guidance for selecting recurrent architectures compatible with EMR;
3. A co-design approach that jointly optimizes efficiency via architectural and inference choices.

### 3.1 Characteristics of Recurrent LLMs and Inference Strategies

**Recurrent LLMs: strengths and limitations.** Recurrent LLMs process each token with constant computation and memory, enabling efficient long-context handling. However, their fixed-size memory limits the amount of information retained at any moment. *The central challenge is the trade-off between efficiency and performance on long-context tasks.*

**Drawbacks of One-Round inference.** With One-Round inference, Recurrent LLMs enjoy $O(L)$ compute and $O(1)$ memory complexity, but processing the entire context in a single pass can overwhelm their limited memory, causing notable performance degradation.

**Advantages of End-to-End Multi-Round inference.** EMR enables incremental, selective processing. By reading in steps, the model can retain only the most relevant information, avoiding memory overload. Although EMR introduces token-generation overhead, yielding $O(\lambda L)$ compute and $O(1)$ memory complexity, it remains far more efficient than the quadratic complexity of Self-Attention LLMs with One-Round inference. Crucially, Recurrent LLMs' constant-size hidden memory can be preserved and updated across rounds at no extra memory cost, which leads to superior performance compared to Self-Attention LLMs using Non-End-to-End Multi-Round inference (which discards hidden states between rounds).

In summary, *Smooth Reading* combines Recurrent LLMs with EMR to preserve efficiency while strengthening long-context understanding. As summarized in Table 1, One-Round inference best

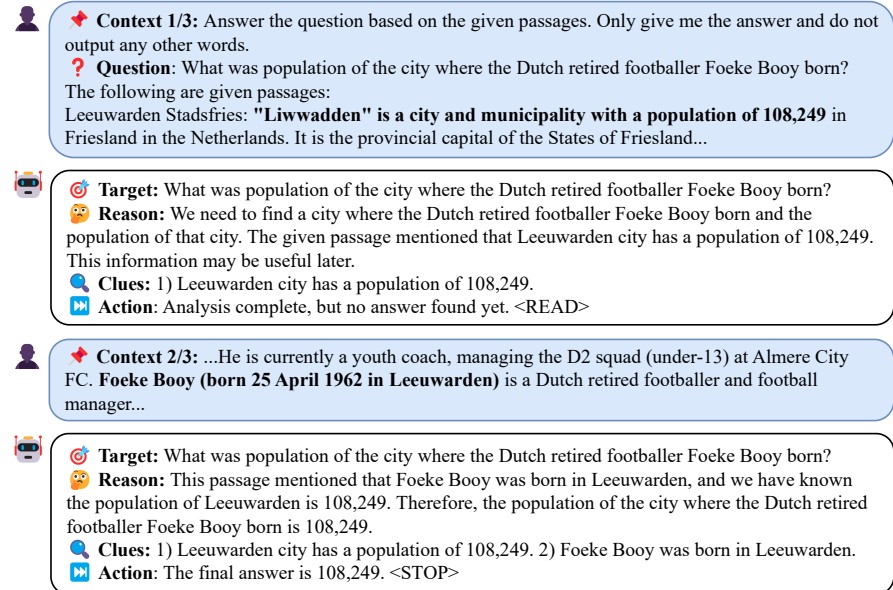

Figure 3: Inference Example of *Smooth Reading*. White boxes indicate the information gathered by the model and the decision points for whether to continue reading. Blue boxes represent chunks of the query context.

suits Self-Attention LLMs, whereas EMR is preferable for recurrent architectures, underscoring the importance of architecture-inference co-design. Recurrent LLMs with EMR also offer the best balance between performance and efficiency among these strategies.

## 3.2 END-TO-END MULTI-ROUND INFERENCE FOR RECURRENT LLMS

**End-to-End Multi-Round agentic pipeline.** Our EMR procedure proceeds in rounds: at each step, the model takes an action (including `<READ>` and `<STOP>`) and receives an observation, iterating until it decides that enough information has been gathered. This stepwise process avoids memory overload and ensures that relevant context remains locally available (see Figure 2c).

**Chunked reading to avoid memory bottlenecks.** To respect recurrent memory limits, we partition the input into manageable, semantically coherent chunks processed sequentially. Each step introduces only a small amount of new information, reducing overflow risk and encouraging focus on salient content. Models advance by emitting a `<READ>` action. Specifically, we use a lightweight, rule-based hierarchical chunker driven by a prioritized delimiter list. At each level, the context is split (e.g., first by paragraphs using "\n"), then adjacent units are merged until a maximum chunk size is reached. If a unit exceeds the limit, it is recursively split using finer-grained delimiters (e.g., sentences via "."), and merging is repeated. In practice, we apply the following delimiters in order: "\n\n\n", "\n\n", "\n", ". ", ". ", "! ", "? ", ", ", "; ", ": ", " – ", and " " (space), yielding size-bounded chunks with reasonable semantic boundaries. To avoid tokenization overhead, we estimate chunk length using $n_{\text{token}} \approx \text{Int}(1.5 \times n_{\text{words}})$, based on the empirical average of $\sim 1.5$ tokens per word.

**Contextual summary compression.** After reading each chunk, the model produces a contextual summary that distills essential information. These summaries (1) represent key content with fewer tokens where possible and (2) update the hidden state so that recent, relevant information is prioritized in recurrent memory. As illustrated in Figure 3, each contextual summary contains: 1) `Target`: the task objective, to prevent distraction from irrelevant context; 2) `Clues`: salient information relevant to solving the task, such as accumulated summaries (for summarization) or query-related facts (for QA); 3) `Reason`: a brief rationale for updating clues at the current reading step. These elements ensure that the model can answer the query using focused, relevant context.

**Preserving hidden memory throughout inference.** Unlike Self-Attention LLMs, Recurrent LLMs maintain a fixed-size hidden memory regardless of context length. EMR retains and incrementally

updates this memory across rounds, accumulating both recent and long-range information without additional memory cost.

**Early stopping mechanism.** After each chunk, the model decides whether it has sufficient information to answer the query. If so, it stops reading and produces the answer with a `<STOP>` action; otherwise, it continues. This adaptive early stopping conserves computation.

An algorithmic summary of our EMR is provided in Algorithm 1 for reference.

### 3.3 ARCHITECTURAL CONSIDERATIONS FOR RECURRENT LLMS

Different inference strategies favor different architectural properties: **One-Round inference** benefits from architectures with maximal effective memory, since all information must be retained simultaneously. **End-to-End Multi-Round inference** (Smooth Reading) relies more on robust length generalization and dynamic compression than on large single-pass memory. The emphasis shifts from raw memory capacity to length extrapolation beyond the training regime.

Two principal recurrent families are: (1) sliding-window attention models (Beltagy et al., 2020), and (2) linear-attention models (Katharopoulos et al., 2020). While linear-attention approaches can offer larger effective memory under One-Round inference (Liu et al., 2025; Yang et al., 2025; Peng et al., 2025), sliding-window attention exhibits stronger length extrapolation under EMR. Our experiments indicate that sliding-window architectures are better suited for Smooth Reading at extreme sequence lengths.

### 3.4 CO-DESIGN: OPTIMIZING EFFICIENCY

Efficiency emerges from the joint choice of architecture and inference. We analyze how EMR maintains linear complexity and how to optimize wall-clock time via co-design. Let the per-token wall-clock times for prefilling and decoding be functions of model complexity $s$: $\mathrm{p}(s)$ and $\mathrm{d}(s)$, respectively. These scale with $s$ and are independent of context length. For inference, define chunk size $c$, number of chunks $n$, and total context length $L = c \times n$. Let $g$ be the average number of decoded tokens per round, assumed independent of $L$. The total inference time is

$$T_{\text{Recurrent-EMR}} \;=\; n \cdot c \cdot \mathrm{p}(s) \;+\; n \cdot g \cdot \mathrm{d}(s) \;=\; L\left(\mathrm{p}(s) + \tfrac{g}{c}\,\mathrm{d}(s)\right). \tag{1}$$

**Linear complexity**: For fixed $s$, $c$, and $g$, $T_{\text{Recurrent-EMR}}$ scales linearly with $L$. **Efficiency optimization**: Efficiency depends jointly on model complexity $s$ and chunk size $c$: wall time increases with $s$ but decreases as $c$ grows (larger chunks reduce the number of rounds and, hence, decoding steps). However, larger $c$ requires greater effective memory. Optimal efficiency therefore requires co-designing architecture and inference method. In practice, we jointly tune the window size (controlling memory span and effective complexity) and the chunk size to optimize sliding-window models under EMR.

### 3.5 DATASET CONSTRUCTION

To train models for *Smooth Reading*, we construct a supervised fine-tuning dataset covering several long-context tasks (e.g., QA). For each task, we:

1. **Collect raw data**: use standard training splits containing query, answer, and context;
2. **Generate summaries**: We generate stepwise contextual summaries and final answers using either a rule-based method or a state-of-the-art LLM (DeepSeek-V3 (et al., 2024)), selected based on task complexity. Rule-based generation is applied to simpler tasks (e.g., retrieval), whereas the LLM is used for more complex tasks (e.g., summarization and question answering). For LLM-based generation, we adopt a Non-End-to-End Multi-Round pipeline for efficiency and use one-shot prompting with manually crafted examples to guide contextual summary generation. The prompt template is shown in Appendix A.1.1. After each contextual summary, we append a `<READ>` token if reading continues; otherwise, we append a `<STOP>` token. Maximum chunk sizes are varied during summary generation to enhance robustness, ranging from 128 to 4096 tokens.

Table 2: Results on LongBench. "Infer" indicates the inference method: "OR" (One-Round), "EMR" (End-to-End Multi-Round), and "NMR" (Non-End-to-End Multi-Round). Best results in **bold**.

| Architecture | Infer | Model | SQA | MQA | Summary | FewShot | Synthetic | Code | Avg |
|---|---|---|---|---|---|---|---|---|---|
| Self-Attention | OR | Qwen-2.5-3B-OR | 24.20 | 41.25 | **30.22** | 65.89 | 56.75 | 66.00 | 47.38 |
| | NMR | Qwen-2.5-3B-NMR | **31.54** | 42.17 | 21.91 | 68.96 | 61.50 | 64.13 | 48.37 |
| Recurrent | OR | RWKV-7-3B-OR | 16.96 | 11.39 | 29.16 | 67.72 | 60.50 | 62.84 | 41.43 |
| | | SWA-3B-4k-OR | 16.43 | 26.14 | 26.96 | 66.43 | 48.00 | **66.22** | 41.70 |
| | EMR | RWKV-7-3B-EMR | 28.87 | 40.02 | 28.23 | 65.90 | 65.25 | 59.92 | 48.03 |
| | | SWA-3B-4k-EMR | 30.46 | **47.67** | 26.27 | **69.60** | **66.75** | 65.18 | **50.99** |

Table 3: Results on the Needle-in-a-Haystack (NIAH) benchmark. All models are trained with a 32k context length and evaluated at context lengths ranging from 8k to 256k.

| Architecture | Infer | Model | 8k | 16k | 32k | Avg | 64k | 128k | 256k | Avg |
|---|---|---|---|---|---|---|---|---|---|---|
| Self-Attention | OR | Qwen-2.5-3B-OR | 98.80 | 98.60 | 97.00 | 98.13 | 98.00 | 26.00 | 0.00 | 41.33 |
| | NMR | Qwen-2.5-3B-NMR | 87.80 | 87.00 | 92.60 | 89.13 | 93.80 | 95.00 | 67.20 | 85.33 |
| Recurrent | OR | RWKV-3B-OR | 98.40 | 95.80 | 86.60 | 93.60 | 39.00 | 8.60 | 0.00 | 15.87 |
| | | SWA-3B-4k-OR | 53.60 | 22.40 | 11.60 | 29.20 | 6.80 | 1.80 | 1.60 | 3.40 |
| | EMR | RWKV-3B-EMR | 99.40 | 98.80 | 97.20 | 98.47 | 75.20 | 20.60 | 1.94 | 32.58 |
| | | SWA-3B-4k-EMR | **99.80** | **100.00** | **100.00** | **99.93** | **100.00** | **99.80** | **99.60** | **99.80** |

3. **Enable early stopping**: for tasks amenable to early stopping (e.g., QA), allow the generator to halt reading once sufficient information has been found;

4. **Clean the data**: apply rule-based filters to remove low-quality outputs.

In total, the dataset comprises 48,856 items. We use this dataset to train models to produce contextual summaries, embedding Smooth Reading behavior directly into the model. The configuration of our dataset construction is summarized in Table 5.

## 4 EXPERIMENTS

### 4.1 EXPERIMENTAL SETUP

We systematically compare model architectures and inference methods to assess the effectiveness of *Smooth Reading*. We evaluate four configurations:

1. Self-Attention LLM with One-Round inference (OR).
2. Self-Attention LLM with Non-End-to-End Multi-Round inference (NMR).
3. Recurrent LLM with One-Round inference (OR).
4. *Smooth Reading (ours)*: Recurrent LLM with End-to-End Multi-Round inference (EMR).

Our experimental pipeline comprises two steps:

**Base Model Preparation:** For Self-Attention LLMs, we use Qwen-2.5 (Qwen et al., 2025) as a representative model. For Recurrent LLMs, we consider: 1) **Sliding-Window LLMs:** Based on Qwen-2.5, modified to use a sliding-window attention mechanism (denoted "SWA-$x$k," where $x$ is the window size; 4k tokens unless otherwise specified). The conversion process is detailed in Appendix A.1.2. 2) **RWKV-7 (Peng et al., 2025):** A strong Recurrent LLM employing a variant of linear attention.

**Model Training and Evaluation:** We derive three dataset variants tailored to each inference method (OR, NMR, EMR), and use them to train all models—including Self-Attention LLM (Qwen-2.5-3B), Sliding-Window LLM (SWA-3B-4k), and RWKV-7 (RWKV-7-3B)—under identical training settings for fair comparison. The inference method is appended to each model name (e.g., Qwen-2.5-3B-OR, SWA-3B-4k-EMR). Further training and evaluation details are provided in Appendix A.1.3 and Appendix A.1.4.

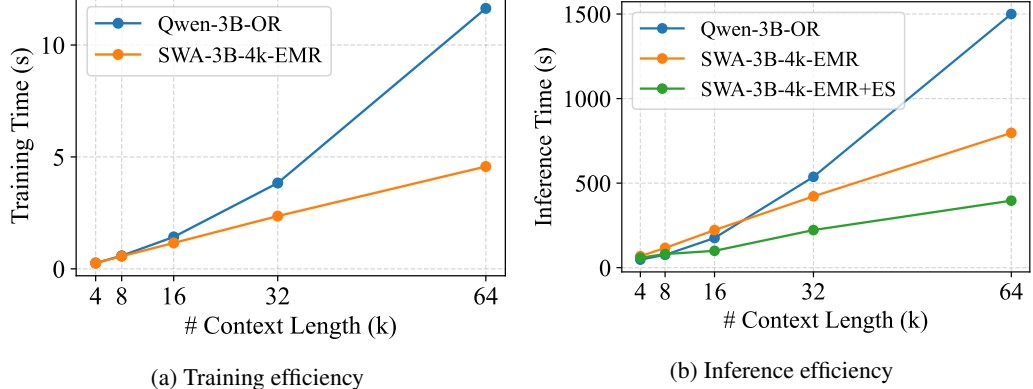

(a) Training efficiency

(b) Inference efficiency

Figure 4: Efficiency vs. Context Length. Recurrent LLMs with *Smooth Reading* scale linearly; Self-Attention LLMs scale quadratically. "+ES" indicates early stopping. The training speed of SWA-3B-4k-EMR is about 2.5x that of Qwen-2.5-3B-OR at 64k. At 64k, SWA-3B-4k-EMR is about 2x faster than Qwen-2.5-3B-OR during inference, improving to up to 4x with early stopping.

## 4.2 LONG-CONTEXT PERFORMANCE COMPARISON

We evaluate on LongBench (Bai et al., 2024) and NIAH (Hsieh et al., 2024) (Tables 2 and 3). For NIAH, we report both in-distribution lengths (8k-32k, matching the 32k training maximum) and extrapolated lengths (64k-256k).

**One-Round Inference:** Recurrent LLMs underperform relative to Self-Attention LLMs. Qwen-2.5-3B-OR averages 47.38% on LongBench; RWKV-7-3B-OR and SWA-3B-4k-OR are below 42%. On NIAH within the 32k training limit, Qwen-2.5-3B-OR reaches 98.13%, outperforming recurrent baselines and highlighting the mismatch between One-Round inference and fixed recurrent memory.

**End-to-End Multi-Round inference (*Smooth Reading*):** EMR substantially improves Recurrent LLMs. On LongBench, RWKV-7-3B-EMR and SWA-3B-4k-EMR reach 48.03% and 50.99%, surpassing Qwen-2.5-3B-OR by 0.65 and 3.61 points, respectively. On NIAH, RWKV-3B-EMR and SWA-3B-4k-EMR match or exceed Qwen-2.5-3B-OR; notably, SWA-3B-4k-EMR maintains near-perfect accuracy up to 256k tokens.

**End-to-End vs. Non-End-to-End Multi-Round:** Recurrent LLMs with EMR outperform Self-Attention LLMs with NMR by preserving hidden states across rounds. On LongBench, SWA-3B-4k-EMR exceeds Qwen-2.5-3B-NMR by 2.62 points. On NIAH, NMR is less stable and generally worse than Recurrent LLMs with EMR.

We conduct additional studies in Appendix A.2:

- Scaling to 7B models (Appendix A.2.1) yields consistent gains with *Smooth Reading*.
- Comparisons with additional Self-Attention LLM baselines (e.g., Llama3.1 (Dubey et al., 2024); Appendix A.2.2) further validate the effectiveness of our method.
- Evaluation on out-of-distribution tasks from RULER (Hsieh et al., 2024) and HELMET (Yen et al., 2025) (Appendix A.2.3), including tasks unseen during training, demonstrates strong generalization.
- Comparisons with additional inference methods—RAG, CompACT (Appendix A.2.5), and OPRM (Appendix A.2.4)—show consistent advantages.

## 4.3 EFFICIENCY COMPARISON

A key advantage of Recurrent LLMs is linear computational complexity in context length, compared with the quadratic cost of Self-Attention LLMs. We report training and inference efficiency using the Sliding-Window LLM; RWKV-7 is omitted from the efficiency plots due to the lack of a highly-optimized inference engine for fair comparison.

Table 4: Ablation study on window size (W) and chunk size (C) for accuracy (%) and inference time.

(a) Accuracy

| W \ C | 512 | 1024 | 2048 | 4096 |
|---|---|---|---|---|
| 512 | 97.0 | 77.8 | 0.0 | 0.0 |
| 1024 | 99.2 | 89.0 | 26.4 | 12.6 |
| 2048 | 100.0 | 100.0 | 95.4 | 31.2 |
| 4096 | 99.8 | 99.8 | 100.0 | 83.4 |

(b) Inference Time (Seconds)

| W \ C | 512 | 1024 | 2048 | 4096 |
|---|---|---|---|---|
| 512 | 528 | 457 | 327 | 100 |
| 1024 | 537 | 444 | 374 | 364 |
| 2048 | 577 | 471 | 387 | 343 |
| 4096 | 646 | 505 | 423 | 366 |

**Training efficiency:** Sliding-Window LLMs train substantially faster than Self-Attention LLMs. As shown in Figure 4a, at a 64k context length, SWA-3B-4k-EMR is $2.5\times$ faster than Qwen-2.5-3B-OR.

**Inference efficiency:** We evaluate inference speed on NIAH. Sliding-Window LLMs with *Smooth Reading* are slightly slower than Self-Attention LLMs at short lengths due to overhead, but scale linearly and become advantageous as the length grows. At 64k, SWA-3B-4k-EMR is $2\times$ faster than Qwen-2.5-3B-OR (Figure 4b).

**Early stopping:** Our method supports halting once sufficient information is found, further reducing latency. At 64k, SWA-3B-4k-EMR+ES improves from $2\times$ faster to up to $4\times$ faster than Qwen-2.5-3B-OR. An ablation in Appendix A.2.6 shows that early stopping does not compromise performance.

**Inference time vs. accuracy:** We compare inference time versus accuracy across architecture-inference combinations in Figure 1. SWA-3B-4k-EMR closes the performance gap to Qwen-2.5-3B-OR with only a 20% increase in inference time over SWA-3B-4k-OR, while remaining $2\times$ faster than Qwen-2.5-3B-OR.

## 5 ANALYSIS AND ABLATION STUDY

### 5.1 LENGTH EXTRAPOLATION AND SCALABILITY

We evaluate length extrapolation up to 256k tokens on the NIAH (Hsieh et al., 2024) benchmark, as shown in Table 3. All models are trained with a 32k context length and tested on substantially longer sequences.

**Self-Attention LLMs:** Under One-Round (OR) inference, models such as Qwen-2.5-3B-OR perform well within the training length but collapse at long contexts (0% at 256k). The Non-End-to-End Multi-Round (NMR) strategy extrapolates better than One-Round (OR), yet still degrades at 256k and is unstable because it does not preserve hidden memory.

**Sliding-Window LLMs:** With One-Round (OR) inference, performance declines as context length grows due to limited memory capacity (e.g., SWA-3B-4k-OR reaches 53.6% at 8k). In contrast, with EMS, scalability is excellent, achieving 99.6% at 256k.

**RWKV:** With One-Round (OR) inference, RWKV is strong within the training range (86.6% at 32k) but degrades rapidly beyond it. With EMS, RWKV-3B-EMR reaches 75.2% at 64k, then declines on longer sequences.

**Comparison of SWA and RWKV:** Figure 5b compares length extrapolation for SWA-3B-4k-OR and RWKV-3B-OR while keeping the needle's relative position fixed. SWA-3B-4k-OR has limited local memory (it can only retain content within its 4k window) but exhibits stable length extrapolation, maintaining performance beyond the training length. RWKV-3B-OR can remember much longer spans within the training range, yet its performance drops sharply beyond that. Thus, One-Round (OR) inference favors RWKV's longer memory, whereas *Smooth Reading* benefits more from SWA's stronger length extrapolation. These results underscore that **different inference methods call for different architectural properties**.

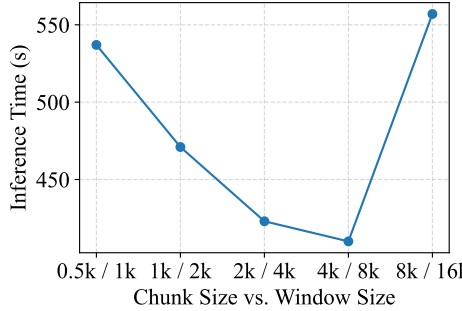 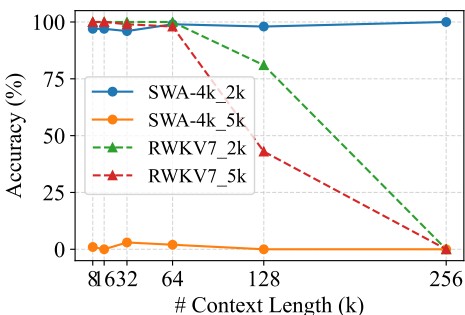

(a) Inference time with a fixed chunk-to-window ratio    (b) Length extrapolation comparison on NIAH

Figure 5: (a) Inference time with a fixed chunk-to-window ratio of 1:2. (b) Length extrapolation comparison of a Sliding-Window LLM (SWA-4k) and RWKV-7 on NIAH. Curves labeled "{arch}_{needle position}" indicate different models and the relative position of the searched token. SWA-4k exhibits stronger length extrapolation, while RWKV-7 provides longer memory within the training context but limited extrapolation.

## 5.2 IMPACT OF CO-DESIGN ON PERFORMANCE AND EFFICIENCY

We ablate the *window size* (model memory span) and *chunk size* (processing granularity) for Sliding-Window LLMs on NIAH. Results are summarized in Table 4.

**Chunk size:** Larger chunks reduce the number of processing rounds and speed up inference, but if the chunk size exceeds the window size, accuracy collapses (e.g., W=512, C=4096 yields 0%). With a sufficient window, increasing the chunk size from 512 to 4096 markedly reduces time (e.g., 646s to 366s at W=4096).

**Window size:** Larger windows retain more information and permit larger chunks, improving accuracy (e.g., at C=2048, accuracy rises from 0% at W=512 to 100% at W=4096), but they increase per-step computation and thus inference time for a fixed chunk size.

**Interaction between chunk and window sizes:** Joint tuning is essential: 1) For high accuracy, the chunk size should not exceed the model's window (hidden memory) size. 2) For efficiency, a moderate ratio (e.g., chunk:window = 1:2) balances fewer rounds against manageable per-step complexity.

Fixing the chunk-to-window ratio at 1:2 and scaling both, inference time follows a U-shaped trend (Figure 5a): very small or very large sizes are inefficient, whereas moderate settings (e.g., W=4096 or W=8192) are fastest. These findings highlight the importance of co-designing architecture and inference: balancing model complexity (window size) and processing steps (chunk size) is critical for both efficiency and performance of Recurrent LLMs on long-context tasks.

## 6 CONCLUSION

Recurrent LLMs have underperformed on long-context tasks primarily because One-Round inference is misaligned with their fixed-size memory. We make two contributions: (i) an End-to-End Multi-Round (EMR) inference strategy that processes long inputs incrementally and substantially improves Recurrent LLMs, demonstrating that better inference can overcome apparent architectural limits; and (ii) evidence that architecture–inference co-design, rather than architecture alone, is critical for accuracy, scalability, and efficiency. Together, these findings reveal the untapped potential of recurrent architectures and establish a co-designed paradigm for long-context language modeling. Last, we discuss limitations and future directions in Appendix A.3.

## ACKNOWLEDGMENTS

This research project was supported by Shanghai Artificial Intelligence Laboratory.

## ETHICS STATEMENT

We declare that there are no conflicts of interest that could inappropriately influence this work. Our study does not involve human subjects, the collection of personal data, or experiments involving protected groups. All datasets used in this work are publicly available and widely adopted in the research community, such as the LongBench and Needle-in-a-Haystack benchmarks. These datasets have been carefully curated to minimize biases and ethical risks.

## REPRODUCIBILITY STATEMENT

We provide comprehensive implementation details in Section 3, Section 4.1, and Appendix A.1, including our method, dataset construction, training, evaluation protocols, and all hyperparameters, to ensure full reproducibility. In addition, we will release our code and datasets upon publication to facilitate independent verification and further research.

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

## A  APPENDIX

---
**Algorithm 1** Smooth Reading: End-to-End Multi-Round Inference

---
**Require:** Context $D$, chunk size $C$, recurrent model $M$, initial hidden state $\mathbf{h}_0 \in \mathbb{R}^d$
**Ensure:** Answer $a$
1: $\mathcal{X} \leftarrow \text{CHUNK}(D, C)$ ▷ Partition context into chunks of size $C$
2: $\mathbf{h} \leftarrow \mathbf{h}_0$ ▷ Initialize hidden state tensor
3: $a \leftarrow \text{NULL}$ ▷ Initialize answer
4: **for** $i = 1$ **to** $|\mathcal{X}|$ **do**
5:     response, $\mathbf{h} \leftarrow M(\mathcal{X}[i], \mathbf{h})$ ▷ Process chunk, get text response and updated state
6:     action, summary $\leftarrow \text{PARSERESPONSE}(\text{response})$ ▷ Extract action and summary from text
7:     **if** action = "READ" **then**
8:         **continue** ▷ Process next chunk
9:     **else if** action = "STOP" **then**
10:         $a \leftarrow \text{PARSERESULT}(\text{summary})$
11:         **break** ▷ Early termination with answer
12:     **end if**
13: **end for**
14: **return** $a$

---

Table 5: Configuration for Dataset Construction. "Num" indicates the number of samples, and "Avg. Length" indicates the average length of the samples. The raw datasets include the training sets of HotpotQA (Yang et al., 2018), NarrativeQA (Kočiský et al., 2017), GovReport (Huang et al., 2021), QMSum (Zhong et al., 2021), TREC (Li & Roth, 2002), TriviaQA (Joshi et al., 2017), SAMSum (Gliwa et al., 2019), WikiSum (Liu et al., 2018), and LCC (Guo et al., 2023). Note, we only use the training sets of these datasets to construct our dataset, to avoid data leakage.

| Task | Raw Dataset | Summary Generator | Early Stop | Clean Metric | Num | Avg. Length |
|---|---|---|---|---|---|---|
| Question Answering | HotpotQA | LLM | Yes | F1 | 5,652 | 4,249 |
| | NarrativeQA | LLM | Yes | F1 | 4,348 | 10,173 |
| Summarization | GovReport | LLM | No | Rouge-L | 8,114 | 12,411 |
| | QMSum | LLM | No | Rouge-L | 742 | 15,245 |
| Few-shot | TREC | LLM | No | Exact Match | 3,259 | 7,759 |
| | TriviaQA | LLM | No | F1 | 3,370 | 13,942 |
| | SAMSum | LLM | No | Rouge-L | 3,371 | 12,100 |
| Passage Count | WikiSum | Rule | No | Exact Match | 3,333 | 15,777 |
| Retrieval | WikiSum | LLM | Yes | Exact Match | 3,333 | 8,254 |
| | HotpotQA | Rule | Yes / No | Exact Match | 1,667 | 20,117 |
| Code Generation | LCC | LLM | No | Edit Similarity | 10,000 | 3,331 |

## A.1 IMPLEMENTATION DETAILS

### A.1.1 MORE DETAILS ABOUT DATASET CONSTRUCTION

Table 5 summarizes the configuration for our dataset construction, and below is the prompt template we use to generate stepwise contextual summaries with LLMs.

> {example}
>
> Above is an example.
>
> Now, you need to address a problem with a long context, chunk by chunk. You can access one chunk at a time and the saved information. You should update the information to solve the problem. The saved and updated information are in JSON format. They have four keys.
>
> - "target": The target of the request. If there is no target, please leave it empty. The output format requirement should be included in the target.
> - "reason": The reason for updating the information. You should explain why you update the information according to the given passage.
> - "clues": The updated information. You should update the information according to the given passage. The format is a list of strings.
> - "result": The result of the request. You should give the answer to the target if the information is enough. If the information is not enough, please leave it empty.
>
> Respond in the format of JSON.
>
> Summary of the previous chunks:
>
> {Summary of Previous Chunks}
>
> Current chunk:
>
> {Chunk Content}

### A.1.2 HOW TO CONVERT SELF-ATTENTION LLM TO SLIDING-WINDOW LLM

It is simple to convert Self-Attention LLMs to Sliding-Window LLMs. As Sliding-Window LLMs do not introduce any new parameters, we directly inherit all parameters from the Self-Attention LLM, such as MLP layers and QKVs of attention layers. Then we modify the self-attention mechanism to attend only to the tokens within the sliding window.

### A.1.3 TRAINING DETAILS

We use AdamW (Loshchilov & Hutter, 2019) with a learning rate of 4e-5 and a weight decay of 0.01. The batch size is set to 2, and the context length is set to 32k. We use a cosine learning rate decay schedule with warmup. As an exception, we use a learning rate of 1e-5 and a weight decay of 0.1 for models with 7B parameters, as we find this setting to be more effective. We train our models with Xtuner (Contributors, 2023b). For models with 3B parameters, we train them for 1 epoch on one H800 GPU, with about 12 hours of training time.

### A.1.4 EVALUATION DETAILS

We evaluate our models using the LongBench (Bai et al., 2024) and NIAH (Hsieh et al., 2024) benchmarks.

**LongBench**: LongBench comprises five main task categories, each with several sub-tasks:

- **Single-document question answering (SQA)**: NarrativeQA (Kočiský et al., 2017), Qasper (Dasigi et al., 2021), MultiFieldQA (Bai et al., 2024)
- **Multi-document question answering (MQA)**: HotpotQA (Yang et al., 2018), 2WikiMultihopQA (Ho et al., 2020), MuSiQue (Trivedi et al., 2022)
- **Summarization**: GovReport (Huang et al., 2021), QMSum (Zhong et al., 2021), MultiNews (Fabbri et al., 2019)

Table 6: Results on LongBench with 7B-parameter Models.

| Architecture | Infer | Model | SQA | MQA | Summary | FewShot | Synthetic | Code | Avg |
|---|---|---|---|---|---|---|---|---|---|
| Self-Attention | OR | Qwen-2.5-7B-OR | 34.90 | **56.72** | **32.07** | **72.77** | 57.75 | **73.41** | **54.60** |
| | NMR | Qwen-2.5-7B-NMR | 34.37 | 45.37 | 25.20 | 71.27 | **66.50** | 65.26 | 51.33 |
| Recurrent | OR | SWA-7B-4k-OR | 26.10 | 37.04 | 28.12 | 71.50 | 42.38 | 71.88 | 46.17 |
| | EMR | SWA-7B-4k-EMR | **37.80** | 54.67 | 27.39 | 69.86 | 65.75 | 67.71 | 53.86 |

Table 7: Results on LongBench with More Models

| Architecture | Infer | Model | SQA | MQA | Summary | FewShot | Synthetic | Code | Avg |
|---|---|---|---|---|---|---|---|---|---|
| Self-Attention | OR | LongChat-v1.5-7B (Li et al., 2023) | 28.7 | 20.6 | 26.7 | 60.0 | 15.8 | 54.1 | 34.3 |
| | | ChatGLM2-6B (GLM et al., 2024) | 32.9 | 33.7 | 27.6 | 59.1 | 39.2 | 52.7 | 40.9 |
| | | Llama-3.1-8B-Instruct (Dubey et al., 2024) | 25.3 | 23.7 | 28.4 | 69.4 | 53.0 | 56.6 | 42.7 |
| | | Mistral-7B-Instruct-v0.3 (Jiang et al., 2023) | **41.3** | 39.0 | 27.2 | 70.7 | 51.8 | 49.1 | 46.5 |
| | | ChatGLM3-6B (GLM et al., 2024) | 40.3 | 46.6 | 29.5 | 68.1 | 50.5 | 56.2 | 48.5 |
| | | Qwen2.5-7B-Instruct Qwen et al. (2025) | 40.9 | 44.1 | 26.6 | 69.1 | 64.6 | 54.5 | 50.0 |
| Self-Attention | OR | Qwen-2.5-7B-OR (ours) | 34.90 | **56.72** | **32.07** | **72.77** | 57.75 | **73.41** | **54.60** |
| Recurrent | EMR | SWA-7B-4k-EMR (ours) | 37.8 | 54.7 | 27.4 | 69.9 | **65.8** | 67.7 | 53.9 |

- **Few-shot learning**: TREC (Li & Roth, 2002), TriviaQA (Joshi et al., 2017), SAMSum (Gliwa et al., 2019)

- **Synthetic tasks**: PassageCount (Bai et al., 2024), PassageRetrieval (Bai et al., 2024)

- **Code generation**: LCC (Guo et al., 2023), RepoBench-P (Liu et al., 2023)

**Needle-in-a-Haystack (NIAH)**: We use essays as haystacks, words as keys, and UUIDs as values. We use only one needle in our experiments by default, but four needles in Table 3, as one needle is insufficient to distinguish between models.

**Chunk Size during Evaluation**: During inference with *Smooth Reading*, we set different chunk sizes according to the model type and task. For Sliding-Window LLMs, we use a chunk size of 1024 for LongBench and 2048 for NIAH. For RWKV-7, we use a chunk size of 512 for LongBench and 256 for NIAH.

**Inference Engine**: We use LMDeploy (Contributors, 2023a), which provides high performance for Self-Attention LLMs and supports Sliding-Window LLMs in interactive mode, maintaining hidden memory throughout inference. It provides strong support for our *Smooth Reading* inference. To support the inference of RWKV-7, we implement a custom inference engine from scratch, as no optimized engine is currently available.

## A.2 MORE EXPERIMENTS

### A.2.1 ADDITIONAL EXPERIMENTS ON 7B MODELS

To further assess the scalability of our method, we conduct experiments using 7B-parameter models, presented in Table 6. SWA-7B-4k-EMR achieves performance comparable to Qwen-2.5-7B-OR with less than a 1% difference and significantly outperforms SWA-7B-4k-OR and Qwen-2.5-7B-NMR. The results confirm that *Smooth Reading* consistently enhances performance across model sizes.

### A.2.2 COMPARISON WITH MORE MODELS

We further compare our models with more commercial and open-source Self-Attention LLMs with 6B-8B parameters, including LongChat-v1.5-7B (Li et al., 2023), ChatGLM2-6B (GLM et al., 2024), Llama-3.1-8B-Instruct (Dubey et al., 2024), Mistral-7B-Instruct-v0.3 (Jiang et al., 2023), ChatGLM3-6B (GLM et al., 2024), and Qwen2.5-7B-Instruct Qwen et al. (2025). As shown in Table 7, our Qwen-2.5-7B-OR is a strong baseline among Self-Attention LLMs, and our SWA-7B-4k-EMR achieves performance comparable to Qwen-2.5-7B-OR, outperforming all other Self-Attention LLMs. Notably, this indicates the solid performance of our SWA-7B-4k-EMR.

Table 8: Evaluation of Generalization Ability on the entire RULER Benchmark. "VT" stands for Variable Tracking, "CWE" for Common Words Extraction, "FWE" for Frequent Words Extraction, and "QA" for Question Answering.

| Model | VT | CWE | FWE | QA | Avg |
|-------|------|------|-------|-------|-------|
| Qwen-2.5-3B-OR | 0.00 | 0.30 | **62.80** | 41.90 | 26.25 |
| Qwen-2.5-3B-NMR | 3.76 | 1.34 | 5.40 | 14.80 | 6.33 |
| SWA-3B-4k-OR | 0.00 | **1.48** | 9.87 | 0.10 | 2.86 |
| SWA-3B-4k-EMR | **8.40** | 0.32 | 50.07 | **53.60** | **28.10** |

Table 9: Evaluation of Generalization Ability on HELMET. "RAG" for Retrieval-Augmented Generation, "PRR" for Passage re-ranking, "ICL" for Many-shot in-context learning, "LQA" for Long-document QA, and "SR" for Synthetic recall.

| Model | RAG | PRR | ICL | LQA | SR | Avg |
|-------|-------|------|-------|-------|-------|-------|
| Qwen-2.5-3B-OR | 47.67 | 0.36 | 34.80 | 14.14 | 31.25 | **25.64** |
| Qwen-2.5-3B-NMR | 21.50 | 0.00 | 2.96 | 12.70 | **35.06** | 14.44 |
| SWA-3B-4k-OR | 10.83 | **0.50** | **39.36** | 15.96 | 7.69 | 14.87 |
| SWA-3B-4k-EMR | **54.44** | 0.47 | 16.68 | **19.51** | 33.94 | 25.01 |

Table 10: Comparison with OPRM on LongBench.

| Infer | Model | SQA | MQA | Summary | FewShot | Synthetic | Code | Avg |
|-------|-------|-------|-------|---------|---------|-----------|-------|-------|
| OPRM | Recurrent-Gemma-IT-9B | 29.83 | 29.60 | 26.45 | 34.23 | 3.75 | 43.41 | 27.88 |
|  | Falcon-Mamba-Instruct-7B | **32.38** | 26.58 | 27.60 | 52.32 | 8.75 | 36.76 | 30.73 |
|  | Falcon3-Mamba-Instruct-7B | 26.91 | 33.86 | 25.57 | 53.05 | 8.75 | 39.93 | 31.35 |
| EMR | RWKV-7-3B-EMR(ours) | 28.87 | 40.02 | **28.23** | 65.90 | 65.25 | 59.92 | 48.03 |
|  | SWA-3B-4k-EMR(ours) | 30.46 | **47.67** | 26.27 | **69.60** | **66.75** | **65.18** | **50.99** |

### A.2.3 MORE EXPERIMENTS ON OUT-OF-DISTRIBUTION BENCHMARKS

To assess the generalization ability of our method, we conduct additional experiments on out-of-distribution (OOD) benchmarks with question formats and prompts unseen during training. We evaluate our model on RULER (Hsieh et al., 2024) and HELMET (Yen et al., 2025) with a 32k context length, and the results are presented in Table 8 and Table 9, respectively. As shown in the tables, our SWA-3B-4k-EMR achieves performance comparable to Qwen-2.5-3B-OR on both benchmarks. Variable Tracking(VT) in RULER and Passage re-ranking(PRR) in HELMET are the tasks that differ most from our training distribution, and both models perform worse on these tasks than on others. Nevertheless, SWA-3B-4k-EMR surpasses Qwen-2.5-3B-OR on these two tasks, indicating stronger OOD generalization. Additionally, both Qwen-2.5-3B-NMR and SWA-3B-4k-OR perform poorly on these benchmarks, further underscoring the importance of co-designing architecture and inference methods.

### A.2.4 COMPARISON WITH OVERFLOW PREVENTION FOR RECURRENT MODELS (OPRM)

Overflow Prevention for Recurrent Models (OPRM; (Ben-Kish et al., 2025)) proposes a RAG-style inference method to address overflow in Recurrent LLMs. Our approach differs in two key ways: (1) OPRM's RAG-style inference constrains generalization and hampers tasks that require reasoning over the entire context; and (2) OPRM does not fully exploit the recurrent architecture and therefore shows no clear advantage over Self-Attention LLMs. We compare our method with OPRM on LongBench (see Table 10), where our model outperforms all OPRM variants by a substantial margin, demonstrating the effectiveness of our approach.

### A.2.5 COMPARISON WITH OTHER MULTI-STEP INFERENCE METHODS

To assess the effectiveness of our approach, we compare *Smooth Reading* with several other LLM inference methods. For a fair evaluation, we implement each method as follows:

Table 11: Comparison with More Multi-Step Inference Methods on Question Answering Tasks.

| Architecture | Infer | Model | HotpotQA | MuSiQue | 2WikiMQA | TriviaQA |
|---|---|---|---|---|---|---|
| Self-Attention | RAG | Qwen-2.5-3B-Instruct | 34.59 | 13.44 | 27.67 | 81.16 |
| | RAG+ | Qwen-2.5-3B-Instruct | 36.91 | 19.92 | 35.47 | 82.15 |
| | CompACT | Qwen-2.5-3B-Instruct | 36.57 | 18.35 | 39.95 | 79.69 |
| Recurrent | EMR | SWA-3B-4k-EMR(ours) | **54.25** | **31.77** | **56.99** | **85.61** |

Table 12: Comparison of Performance with and without Early Stopping on NIAH.

| Early Stop | 8k | 16k | 32k | 64k | 128k | 256k | Avg |
|---|---|---|---|---|---|---|---|
| without | 99.80 | 100.00 | 100.00 | 99.80 | 99.60 | 100.00 | 99.87 |
| with | 99.60 | 99.60 | 99.80 | 99.80 | 99.60 | 99.80 | 99.70 |

- **RAG**: We utilize standard RAG models as introduced by (Asai et al., 2023). For each query, Contriever-MS MARCO retrieves the top five documents from Wikipedia, using the official embeddings from the 2018 English Wikipedia. These retrieved passages form the long context. Note that this method incorporates external knowledge.

- **RAG+**: To avoid using external knowledge, we split the context into multiple passages for each question. Both the query and each passage are encoded using sentence embedding models (Reimers & Gurevych, 2019). We compute the cosine similarity between the query and the passages, select the top three most relevant passages, and provide them as the context for answer generation.

- **CompACT** (Yoon et al., 2024): We also compare with CompACT, a strong baseline. This method iteratively compresses the context based on the question, and the answer is generated from this compressed context. In our experiments, we adopt the off-the-shelf CompACT compressor.[1]

As shown in Table 11, our *Smooth Reading* approach consistently achieves the highest F1-scores across all evaluated datasets, demonstrating its superior capability in handling long-context passages. Compared to the compressor-based method CompACT, *Smooth Reading* attains better results, which suggests that the compression process in CompACT may discard information important for accurate reasoning.

### A.2.6 INFLUENCE OF EARLY STOPPING ON PERFORMANCE

We further compare the performance of our method with and without early stopping by evaluating SWA-3B-4k-EMR on NIAH, as presented in Table 12. The results show that early stopping has minimal impact on performance, with average accuracy exceeding 99% in both scenarios. Moreover, by default, we enable early stopping for question answering on LongBench, and the high performance of our LongBench results (Table 2) also indicates that early stopping does not significantly affect performance.

### A.3 LIMITATION AND FUTURE WORK

There are several limitations to our work.

**No Completely New Architecture or Inference Method** We focus on the *co-design of architecture and inference* rather than introducing entirely new architectures or inference algorithms. Although the individual components we build upon already exist, our contributions are twofold: (1) to the best of our knowledge, we are the first to systematically study the interaction between architecture and inference method, demonstrating that judicious co-design can substantially improve performance; and (2) we significantly improve the long-context performance of Recurrent LLMs, bringing them on par with Self-Attention LLMs—a notable milestone for Recurrent LLMs. Designing entirely new architectures or inference methods is outside the scope of this paper and is left to future work.

---

[1]`https://huggingface.co/cwyoon99/CompAct-7b`

**Architectural Limitations** We study sliding-window LLMs and RWKV as representative recurrent LLM architectures. Other recurrent architectures exist (e.g., DeltaNet (Yang et al., 2025)), but we did not include them due to two practical barriers: (1) the absence of open-source, well-pretrained base models, and (2) the lack of optimized inference engines that support high-throughput, multi-round processing. Extending our approach to these architectures is an important direction for future work.

**Need for SFT Training** Our method requires supervised fine-tuning (SFT) to adapt models to the *Smooth Reading* inference paradigm. This reflects a broader limitation of multi-round inference: current LLMs are mainly trained under a one-round SFT protocol, making their out-of-the-box multi-round performance suboptimal. Prior work on multi-round inference (Yoon et al., 2024; Yu et al., 2025; Shao et al., 2024) similarly relies on additional training. As future work, we plan to explore training regimes that produce models capable of both one-round and multi-round inference without requiring separate SFT datasets.

**Need for Data Construction** Because SFT is required, we must construct training data tailored to multi-round inference. Advances in reinforcement learning (RL) may reduce this requirement by leveraging simpler annotations, thereby lowering data construction costs. Given the complexity of RL and the current lack of mature RL infrastructure for Recurrent LLMs, we leave this to future work.

**Order of Query and Context**

We use the default order of query followed by context (i.e., "`[QUERY] [CONTEXT]`") and do not support the reversed order ("`[CONTEXT] [QUERY]`"). This limitation is common among Multi-Round Inference methods, as most prior work (Yoon et al., 2024; Yu et al., 2025) also adopts this order. Nevertheless, it is addressable: since our agentic multi-step pipeline has an expandable action space, future extensions could allow the model to read chunks in any order and learn to select the reading sequence autonomously. This would enable the model to prioritize relevant chunks regardless of query position. Achieving this, however, would require more sophisticated data construction and training strategies, such as reinforcement learning, which we leave for future work.

**Limited Tasks** We evaluate primarily on long-context tasks, as these pose a central challenge for Recurrent LLMs. While our method is, in principle, applicable to other domains such as deep research (Zheng et al., 2025) and software development (Jimenez et al., 2024), a broader evaluation is left to future work. We expect our approach to improve both accuracy and efficiency in these settings.

Despite these limitations, we believe our work takes a meaningful step toward unlocking the potential of Recurrent LLMs and advancing the frontier of LLM research.

