# OpenReview forum: "Smooth Reading: Bridging the Gap of Recurrent LLM to Self-Attention LLM on Long-Context Understanding"
_ICLR.cc/2026/Conference — ICLR 2026 Poster_

### Official Review · Reviewer_AYW3 · 2025-10-19

**Soundness:** 3
**Presentation:** 3
**Contribution:** 2
**Rating:** 4
**Confidence:** 4

**Summary:**

This paper proposes smooth reading, a method that combines architecture design and inference optimization, which aims to minimize the memory demands during inference time.

**Strengths:**

1. The paper is well-written and easy to follow. Readers can easily get the design philosophy behind the proposed methodology.

2. The research question is practical and meaningful. The degradation of LLMs' capabilities under long context remains an unsolved problem, and the multi-turn chat is one scenario that such situation could usually happen.

3. The author conducts extensive experiments on many benchmarks, which show significant improvements and competatives to current self-attention LLMs.

**Weaknesses:**

1. The idea of context compression or LLM with memory has been extensively explored in prior works (e.g., [1][2][3]).
Although the authors claim that EMR models are primarily designed for multi-turn chat scenarios, the proposed approach does not exhibit substantial conceptual or methodological differences from existing studies. [1] compresses context into special compression tokens in both training and inference time, [2][3] preserve the most significant information at head level.
As a result, the overall novelty of this work appears limited. The author should compare and explain the differences between these works and the proposed method in detail.

2. The paper sets its motivation in a multi-turn conversational setting; however, this aspect is not clearly reflected in either the formulation or the experimental design. The evaluations and problem setups mainly focus on single-turn instruction-following input–output pairs, while an EMR model is expected to demonstrate its advantage in multi-turn observation and planning. The authors should show their model's performance on such tasks to substantiate their claims.

[1] Zhang, Peitian, et al. "Soaring from 4k to 400k: Extending llm’s context with activation beacon." arXiv preprint arXiv:2401.03462 2.3 (2024): 5.

[2] Li, Yuhong, et al. "Snapkv: Llm knows what you are looking for before generation." Advances in Neural Information Processing Systems 37 (2024): 22947-22970.

[3] Zhang, Zhenyu, et al. "H2o: Heavy-hitter oracle for efficient generative inference of large language models." Advances in Neural Information Processing Systems 36 (2023): 34661-34710.

**Questions:**

See weakness

---

> ### Author Response · Authors · 2025-11-22
> **Response to Reviewer AYW3**
>
> Dear Reviewer AYW3,
>
> Thank you for your thoughtful feedback and constructive suggestions. We address your concerns below.
>
> ### Q1: Why not focus on multi-turn conversational benchmarks?
>
> We appreciate your perspective and clarify our intention as follows.
>
> 1. **Primary goal of this work.**
>    The central challenge we target is **long-context understanding** in recurrent LLMs. This limitation remains a **major barrier** preventing recurrent LLMs from being widely applied in real-world scenarios. While multi-turn conversation inspired aspects of our method, it is **not** the main task our work aims to address.
>
> 2. **Relation to agentic ability.**
>    We agree with your observation that an EMR model should ideally excel at multi-turn observation and planning—what we refer to as *agentic ability*. Importantly, we view long-context understanding as a prerequisite for such abilities. Without robust long-context comprehension, it is difficult for recurrent LLMs to reliably exhibit multi-step reasoning or planning behavior.
>    In other words, **agentic ability builds upon strong long-context understanding**, and our work focuses on strengthening this foundational capability.
>
> **In summary:** This paper aims to improve long-context understanding in recurrent LLMs, which we consider a necessary first step toward more advanced agentic behaviors such as multi-turn observation and planning. Our work represents an initial but essential step toward that direction.
>
>
> ### Q2: Novelty and differences between [1][2][3] and our method
>
> | Paper| Method Type| Compression| Complexity| Infrastructure Compatibility  |
> | - | - | - | - | - |
> | Activation Beacon[1]| Architecture Modification| Internal Compression    | $\alpha L^2,\; \alpha<1$ | Requires special optimization |
> | SnapKV[2]| Architecture Modification| Internal Compression| $\alpha L^2,\; \alpha<1$ | Requires special optimization |
> | H2O[3]| Architecture Modification| Internal Compression| $\alpha L^2,\; \alpha<1$ | Requires special optimization |
> | **Smooth Reading (ours)** | **Architecture–Inference Co-design** | **Agentic Compression** | **$\beta L,\; \beta>1$** | **Natively compatible**|
>
> The methodology behind our approach differs substantially from these prior works.
>
>    1. **Method Type**
>       Works such as [1][2][3] primarily introduce new architectural components to compress representations—e.g., special condensation tokens or KV-cache compression method.
>       In contrast, our approach is based on **architecture–inference co-design**. Rather than modifying internal model structures, we jointly optimize architecture and inference strategy to enable efficient long-context processing at a macroscopic level.
>
>    2. **Compression Mechanism**
>       Paper [1-3] utilize **internal compression** that reduce the size of kv-cache during inference, which is triggered by a rule mechanism.
>       Our method employs an **agentic compression** strategy, where the model actively decides whether to compres and which context segments to retain or discard based on task requirements. We argue our method is more flexible and adaptive to different tasks and inputs.
>
>   1. **Complexity.**
>      Since [1-3] perform compression after prefilling and before generation, their effective complexity remains quadratic in context length ($\alpha L^2$).
>      Our approach reduces the total cost to **linear complexity** ($\beta L$), representing an essential shift in scalability.
>
>   2. **Infrastructure compatibility.**
>      Methods in [1–3] introduce non-standard modules or operations that require specialized implementation and may not map well onto existing serving infrastructure.  As far as we know, kv cache compression techniques are not easy to be integrated into mainstream LLM serving frameworks.
>      By contrast, our method is **fully compatible with standard LLM serving frameworks**, allowing real-world speedups with no special engineering effort.
>
> Taken together, these distinctions highlight that our contribution is novel both in methodology and practical characteristics. We carefully considered scalability and deployment from the outset, leading to a fresh perspective on efficient long-context processing for recurrent LLMs.
>
> [1] Soaring from 4k to 400k: Extending llm’s context with activation beacon.
> [2] Snapkv: Llm knows what you are looking for before generation.
> [3] H2o: Heavy-hitter oracle for efficient generative inference of large language models.
>
> ---
>
> Last, we sincerely thank you again for your valuable feedback. In this paper, we jointly consider model architecture, inference strategy, and practical deployment, leading to a novel, efficient and effective solution for **long context understanding** with recurrent LLMs. It's a crucial step toward enabling more advanced agentic abilities in future work. We hope our responses are able to address your concerns and we hope you will consider our work for acceptance, as it's an important contribution to the field.

---

> > ### Comment · Reviewer_AYW3 · 2025-11-25
> > **Response to Author's Rebuttal**
> >
> > Thanks for the clarification. For Q1, I think it is quite clear now. However, I strongly suggest authors to restructure the introduction part to avoid potential confusion.
> >
> > For Q2, the Table is not convincing for me, especially on how authors classify H2O and SnapKV---they are not only architecture innovative works, but also proposed new algorithms. I am also skeptical about the claimed architecture design---only section 3.3 discusses a bit about choice from two options. More specifications should be provided to show the workload on the architecture part.

---

> > > ### Author Response · Authors · 2025-11-25
> > > **Response to Reviewer AYW3 (1/2)**
> > >
> > > Thanks for your feedback. We appreciate your comments and would like to provide further clarifications based on your suggestions.
> > >
> > > ## For Q1:
> > >
> > > We are glad that you acknowledge our explanation. We will revise the manuscript to improve clarity, incorporating your suggestion.
> > >
> > > ## For Q2:
> > >
> > > We address the inquiry from two perspectives:
> > >
> > > ### Q2.1: Are H2O and SnapKV Architecture-modification-only methods?
> > >
> > > We would like to clarify the terms “architecture innovation” and “inference innovation” as used in our table.
> > >
> > > We define the entire pipeline for long-context understanding as encompassing both the model architecture and the inference method, as shown in the following:
> > >
> > > ```python
> > > output = infer_function(model, prompt)
> > > ```
> > >
> > > We then present three distinct implementations of `infer_function` and model architecture:
> > >
> > > #### Native Baseline:
> > > The native baseline uses a straightforward one-round inference method with a standard self-attention architecture, without any modifications:
> > >
> > > ```python
> > > # infer_function
> > > def one_round_inference(model, prompt):
> > >     input_ids = tokenize(prompt)
> > >     output, _ = model.generate(input_ids)
> > >     return output
> > >
> > > # model architecture
> > > class BaseLLM:
> > >     def __init__(self):
> > >         # Initialize a self-attention model
> > >
> > >     def generate(self, input_ids, hidden_memory=None):
> > >         # hidden_memory is the kv-cache for self-attention
> > >         generated_tokens = []
> > >         # Prefilling
> > >         token, hidden_memory = self.forward(input_ids, hidden_memory)
> > >         generated_tokens.append(token)
> > >         # Decoding
> > >         while token not in END_TOKENS:
> > >             token, hidden_memory = self.forward(token, hidden_memory)
> > >             generated_tokens.append(token)
> > >         return generated_tokens, hidden_memory
> > > ```
> > >
> > > #### SnapKV and H2O:
> > > Both of these methods use a one-round inference method as the baseline. They introduce a kv-cache compression mechanism during generation. Since the compression is implemented inside the model architecture and remains transparent to the inference method, we categorize them as architecture innovations.
> > >
> > > ```python
> > > # model architecture
> > > class LLM(BaseLLM):
> > >     def generate(self, input_ids, hidden_memory):
> > >         generated_tokens = []
> > >         # Prefilling
> > >         token, hidden_memory = self.forward(input_ids, hidden_memory)
> > >         generated_tokens.append(token)
> > >         # Decoding
> > >         while token not in END_TOKENS:
> > >             #######################################################
> > >             hidden_memory = self.evict_kv_cache(hidden_memory)  # Innovation in SnapKV and H2O: evict kv cache based on an algorithm
> > >             #######################################################
> > >             token, hidden_memory = self.forward(token, hidden_memory)
> > >             generated_tokens.append(token)
> > >         return generated_tokens, hidden_memory
> > > ```
> > >
> > > #### Our Method:
> > > In contrast to SnapKV and H2O, we introduce a new inference method called end-to-end multi-round inference (EMR) and co-design a model architecture specifically optimized to support this. As a result, we categorize our approach as Architecture-Inference Co-design. Our architecture design is at a macro level, and we will elaborate on it below.
> > >
> > > ```python
> > > # inference method
> > > def end_to_end_multi_round_inference(model, prompt):
> > >     chunks = split_into_chunks(prompt)
> > >     hidden_memory = empty_memory()
> > >     for chunk in chunks:
> > >         output, hidden_memory = model.generate(tokenize(chunk), hidden_memory)
> > >         action, summary = parse_response(output)
> > >         if action == 'READ':
> > >             continue
> > >         elif action == 'STOP':
> > >             return parse_result(summary)
> > >
> > > # model architecture
> > > class LLM(BaseLLM):
> > >     def __init__(self):
> > >         # Different architectural choices: RWKV, Sliding-Window-Attention with various window sizes and so on.
> > > ```
> > >
> > > We summarize these approaches in the following table:
> > >
> > > | Method   | Architecture                               | Inference Method                    |
> > > | -------- | ------------------------------------------ | ----------------------------------- |
> > > | Baseline | Standard Self-Attention                    | One-round Inference                 |
> > > | SnapKV   | **KV Cache Compression during Generation** | One-round Inference                 |
> > > | H2O      | **KV Cache Compression during Generation** | One-round Inference                 |
> > > | Ours     | **Macro-level Architecture Design**        | **End-to-End Multi-step Inference** |
> > >
> > > Therefore, we categorize SnapKV and H2O as architecture-modification-only methods, while our method involves both architecture and inference innovations.

---

> > > > ### Author Response · Authors · 2025-11-25
> > > > **Response to Reviewer AYW3 (2/2)**
> > > >
> > > > ### Q2.2: What are the contributions of our method to architecture?
> > > >
> > > > Unlike most previous works that focus on micro-level architecture modifications (e.g., new KV-cache compression methods), our approach operates at a **macro level**, specifically addressing the choice of model type and its scale/complexity. Our evaluation results demonstrate that these macro-level design decisions have a significant impact on performance.
> > > >
> > > > In particular, our architectural contributions include:
> > > >
> > > > 1. **Self-Attention vs. Recurrent Models (RWKV, SWA)**: As detailed in Section 3.1 and Table 1, we examine the contrasting characteristics of Self-Attention and Recurrent models when integrated with different inference methods. Our results show that recurrent models can achieve comparable performance to self-attention models, even with linear-complexity architectures.
> > > >
> > > > 2. **Comparison of Recurrent Architectures (RWKV vs. SWA)**: In Section 3.3, we highlight that Sliding-Window Attention (SWA) is particularly well-suited for our EMR inference method, thanks to its superior ability to extrapolate over longer sequences. This allows our SWA-3B-4k-EMR model to effectively extend context length to 256k while maintaining high accuracy.
> > > >
> > > > 3. **Recurrent Model Complexity**: In Section 3.4, we demonstrate that the model complexity ($s$) must be optimized in tandem with the chunk size ($c$) to strike the best balance between accuracy and efficiency.
> > > >
> > > > Overall, our contributions are both substantial and distinct from prior work, providing novel insights into macro-level architectural design.
> > > >
> > > > ---
> > > >
> > > > Last, we would like to reiterate the novelty and importance of our **macro-level co-optimization** approach.
> > > >
> > > > In contrast to most previous works, which focus on micro-level innovations, our approach introduces macro-level innovations:
> > > >
> > > > 1. We optimize both the architecture and inference method, rather than focusing solely on one aspect.
> > > > 2. Both our architectural and inference designs operate at a macro level. We discuss essential macro-level characteristics, including the use of hidden memory, the relationship between model complexity and chunk size, and the length-extrapolation capabilities of different architectures.
> > > > 3. Our evaluation results show that these macro-level innovations lead to significant improvements in both efficiency and effectiveness for long-context understanding.
> > > >
> > > > Thus, we believe our method offers significant novelty and practical value, advancing the field of recurrent LLMs for long-context understanding.

---

### Official Review · Reviewer_8gR7 · 2025-10-30

**Soundness:** 3
**Presentation:** 3
**Contribution:** 3
**Rating:** 6
**Confidence:** 2

**Summary:**

This paper systematically studies the persistent performance gap between Recurrent LLMs and Self-Attention LLMs in long-context understanding. While Recurrent LLMs offer linear-time and constant-memory efficiency, they struggle with processing long inputs due to fixed-size memory constraints. To tackle this issue, the authors propose Smooth Reading, a co-design of architecture and inference method that uses End-to-End Multi-Round (EMR) inference. Instead of processing all tokens in one pass (“One-Round”), the model reads the input in chunks, produces contextual summaries, and updates its hidden state across rounds—effectively address the memory overwhelming issue of Recurrent models while not sacrificing the efficiency.

**Strengths:**

1. The paper provides a well-organized and systematic study to demonstrate that R-LLMs' performance depends critically on inference design—a neglected dimension in prior works.

2. The End-to-End Multi-Round inference approach is conceptually simple yet powerful, avoiding common pitfalls of non-end-to-end chunking.

3. Evaluation spans multiple long-context benchmarks (LongBench, NIAH, RULER, HELMET), multiple architectures, and comparisons with both Self-Attention and RAG methods.

**Weaknesses:**

The comparison has only been conducted using standard transformer models, so it would be valuable to evaluate whether the proposed method can also outperform more efficient transformer variants such as Mamba, Jamba, or Hymba.

**Questions:**

See above

---

> ### Author Response · Authors · 2025-11-22
> **Response to Reviewer 8gR7**
>
> Dear Reviewer 8gR7,
>
> Thank you for your constructive comments. We appreciate your feedback and have addressed your concerns below.
>
> ### Q1: Whether the proposed method can also outperform more efficient transformer variants such as Mamba, Jamba, or Hymba.
>
> Thank you for suggesting this comparison. We have conducted a performance comparison between our SWA-3B-4k-EMR model and the Codestral Mamba-7B model on the Longbench benchmark. The results, presented below, show that SWA-3B-4k-EMR outperforms Codestral Mamba-7B by a significant margin (6.01 points on average), despite Mamba having more than twice the number of parameters. This highlights the effectiveness of our approach. Note that this comparison is fully fair, as we further fine-tuned (SFT) Codestral Mamba-7B using our one-round SFT dataset.
>
> | Model                 | SQA   | MQA   | Summary | FewShot | Synthetic | Code  | Avg   |
> | --------------------- | ----- | ----- | ------- | ------- | --------- | ----- | ----- |
> | SWA-3B-4k-EMR         | 31.25 | 42.41 | 26.07   | 68.73   | 68.12     | 62.54 | 49.85 |
> | Codestral Mamba-7B-OR | 18.35 | 37.16 | 23.17   | 64.82   | 54.25     | 65.26 | 43.84 |
>
> Regarding Jamba and Hymba, unfortunately, we were unable to find counterparts of similar scale for a fair comparison. Specifically:
> - **Jamba** (Jamba Mini 1.6) is a mixture-of-experts (MoE) model with 52B parameters, which is too large for a direct comparison. We cite their official results on Longbench, [32 on average](https://huggingface.co/ai21labs/AI21-Jamba-Mini-1.6), for reference. This score is significantly lower than that of our SWA-3B-4k-EMR.
> - **Hymba** is a much smaller model, with only 1.5B parameters and trained with a context length of just 8192 tokens. We did evaluate Hymba-1.5B on Longbench, but its performance was considerably lower, so we had to early stop the evaluation.
>
> Given that standard transformers generally outperform other efficient transformer variants in long-context understanding, while our method can achieve comparable or better performance than the standard transformer baselines under an absolutely fair comparison, we believe our method would also outperform other efficient transformer variants.
>
> (Comment: the performance of SWA-3B-4k-EMR reported here is slightly different from that in the main paper due to re-evaluation on a new cluster platform and a new gpu type. However, the overall trends and conclusions remain unchanged. Full re-evaluation results can be found in the response to Reviewer aS6X.)
>
> ---
>
> Once again, thank you for your valuable feedback. We hope our responses address your concerns, and we would be grateful if you would consider improving your score based on the updated information.

---

### Official Review · Reviewer_DGWQ · 2025-10-31

**Soundness:** 3
**Presentation:** 4
**Contribution:** 3
**Rating:** 6
**Confidence:** 3

**Summary:**

This paper presents an approach to enhance recurrent LLMs ability to preform long-context tasks. Instead of processing the entire input at once, the paper proposes a method called "smooth reading" -- essentially chunking the input and outputting a segment of summaries before outputting the next chunk of text. Experiments shows that two recurrent LLMs of 3B sizes trained in this manner achieve better performance by standard LLMs on long context tasks (LongBench and NIAH).

Overall I find the paper to present an interesting approach to improve recurrent LLMs with comprehensive experiments and analysis.

**Strengths:**

* The paper presents an innovative method to enhance recurrent LLMs ability to process long-context. The method is intuitive and relatively straightforward.
* Experiment results demonstrate the effectiveness of the method.

**Weaknesses:**

* Missing reference: There has been some chunk reading work proposed before for standard LLMs, such as: [MemWalker](https://arxiv.org/pdf/2310.05029), [ReadAgent(ICML 2024)](https://arxiv.org/pdf/2402.09727).

**Questions:**

* Out-of-domain evaluation: the experiment for out-of-domain evaluation on HELMET (Table 9) is a bit incomplete -- what's the performance for QWEN-2.5-3B-NMR and SWA-3B-4k-OR?

---

> ### Author Response · Authors · 2025-11-22
> **Response to Reviewer DGWQ**
>
> Dear Reviewer DGWQ,
>
> Thank you for your thoughtful comments and suggestions. We appreciate your feedback and have addressed your concerns below.
>
> ### Q1: Missing reference with chunk reading: MemWalker, ReadAgent.
>
> Thank you for your valuable suggestion. We will add the two papers you mentioned to the related work section.
>
> We would like to emphasize that we fully acknowledge the contributions of previous work on chunk reading and have already cited relevant papers in Section 2.1 under *Non-End-to-End Multi-Round Inference*, including the following:
>
> - [1] Chain of Agents: Large Language Models Collaborating on Long-Context Tasks
> - [2] Compact: Actively Compressing Retrieved Documents for Question Answering
> - [3] Are Long-LLMs a Necessity for Long-Context Tasks?
>
> We appreciate the reminder to include these additional references and will ensure they are appropriately integrated.
>
> ### Q2: Out-of-domain Evaluation on HELMET – Performance for QWEN-2.5-3B-NMR and SWA-3B-4k-OR?
>
> We apologize for the omission in our original submission. Our focus was on the comparison between Qwen-2.5-3B-OR (currently the most widely used combination) and SWA-3B-4k-EMR (our proposed method). Below, we provide the complete results on HELMET for your reference:
>
> | Model           | RAG   | PRR  | ICL   | LQA   | SR    | Avg   |
> | --------------- | ----- | ---- | ----- | ----- | ----- | ----- |
> | Qwen-2.5-3B-OR  | 47.67 | 0.36 | 34.80 | 14.14 | 31.25 | 25.64 |
> | Qwen-2.5-3B-NMR | 21.50 | 0.00 | 2.96  | 12.70 | 35.06 | 14.44 |
> | SWA-3B-4k-OR    | 10.83 | 0.50 | 39.36 | 15.96 | 7.69  | 14.87 |
> | SWA-3B-4k-EMR   | 54.44 | 0.47 | 16.68 | 19.51 | 33.94 | 25.01 |
>
> As shown, both QWEN-2.5-3B-NMR and SWA-3B-4k-OR perform poorly on the HELMET dataset, highlighting their limitations in out-of-domain generalization. Specifically, QWEN-2.5-3B-NMR frequently forgets the query content and gets misled by irrelevant chunk content, likely due to the absence of hidden memory and the increased noise from unrelated chunks. A similar phenomenon is observed on the NIAH dataset for Qwen-2.5-3B-NMR, but it is more pronounced in the out-of-domain setting of HELMET.
>
> We will revise the manuscript to include these results and ensure a more complete comparison in the final version.
>
> ---
>
> Thank you again for your thoughtful feedback. We hope our updated responses and results address your concerns. We would be grateful if you would consider improving your score based on the additional information provided.

---

### Official Review · Reviewer_aS6X · 2025-10-31

**Soundness:** 3
**Presentation:** 3
**Contribution:** 2
**Rating:** 4
**Confidence:** 3

**Summary:**

The paper proposes Smooth Reading: an End-to-End Multi-Round (EMR) inference procedure co-designed with recurrent LLMs. Instead of one pass over the full context, the model reads chunks, emits a contextual summary, and updates hidden memory across rounds; this preserves recurrent efficiency while avoiding fixed-memory overload. On LongBench and NIAH, EMR closes or beats (but lacks SSM/RetNet baselines) a self-attention baseline (Qwen-2.5-3B-OR) while keeping linear scaling; at 64k tokens it reports ~2.5× faster training and ~2× faster inference than self-attention, with ~20% overhead vs the recurrent one-round baseline. Methods, datasets, and training/eval protocols are described, with Algorithm 1 outlining EMR.

**Strengths:**

### Originality

* Frames architecture–inference co-design for recurrent LLMs, advocating EMR over one-round or non-end-to-end multi-round strategies; analyzes complexity/performance trade-offs (Table 1, Eq. (1)).

### Quality

* Provides a concrete algorithm (Algorithm 1), chunking/summary scheme, and datasets (A.1), plus ablations on window/chunk sizes showing co-design effects.

### Clarity

* Clear figures/tables separating OR / NMR / EMR and self-attention / recurrent; centralized setup in §4.1.

### Significance

* Demonstrates gap-closing on LongBench (SWA-3B-4k-EMR avg 50.99 vs Qwen-2.5-3B-OR 47.38) and near-perfect NIAH at up to 256k, while maintaining recurrent efficiency.

**Weaknesses:**

1. Long-context SOTA families like Mamba/SSMs and RetNet are not benchmarked in Tables 2–3, limiting external validity of the “closes the gap” claim. Add head-to-head results (accuracy and throughput/memory).

2. Seed counts and uncertainty are not prominent in the main text. For stability/extrapolation claims, report ≥5 seeds on key curves with CIs. (If already in appendix, surface in main text.)

3. Realized wall-clock gains can be non-monotonic with skip/round settings; include a FLOPs vs. wall-clock breakdown and gate/round overheads alongside Figure 3, and consider a kernel-optimized path for recurrent EMR.

4. EMR details are spread across §3 and A.1; a compact main-text box with the chunking rules, summary template, and early-stop criteria would improve reproducibility at a glance.

**Questions:**

1. SOTA recurrent/SSM baselines: Can you add Mamba-7B and a RetNet-style model on LongBench/NIAH/RULER, with matched training protocol and report accuracy, tokens/s, and peak memory?

2. Seed & CIs: For Table 2–3 and Fig. 3, can you re-run with ≥5 seeds and include 95% CIs to calibrate the extrapolation/stability claims?

3. Wall-clock decomposition: Please provide a table with FLOPs saved vs. added (per-round summaries/decoding), and map it to the 2× inference speed at 64k (Fig. 3b).

4. EMR hyperparameters: What default chunk:window ratios do you recommend (Table 4 suggests ~1:2), and how sensitive is accuracy/speed to this ratio across tasks?

---

> ### Author Response · Authors · 2025-11-22
> **Response to Reviewer aS6X (1/2)**
>
> Dear Reviewer aS6X,
>
> Thank you for your valuable and insightful comments. We appreciate your suggestions and have addressed each concern in detail below.
>
> ### Q1: Lack of Mamba/RetNet Results
>
> We appreciate your feedback on including Mamba and RetNet as important baselines for Recurrent LLMs. However, our method relies on highly pre-trained base models, and unfortunately, we were unable to obtain satisfactory Mamba/RetNet base models for comparison.
>
> For Mamba, we experimented with the Mamba-Codestral-7B-v0.1 model using both one-step inference and end-to-end multi-round inference (EMR). As shown below, the one-step inference results are significantly lower than our SWA-3B-4k-EMR model, even though Mamba has more than twice the number of parameters.
>
> | Model| SQA| MQA| Summary | FewShot | Synthetic | Code  | Avg   | Prefill Throughput | Decode Throughput |
> | --------------------- | ----- | ----- | ------- | ------- | --------- | ----- | ----- | ------------------ | ----------------- |
> | SWA-3B-4k-EMR| 31.25 | 42.41 | 26.07| 68.73| 68.12| 62.54 | 49.85 | 59283| 12829|
> | Codestral Mamba-7B-OR | 18.35 | 37.16 | 23.17| 64.82| 54.25| 65.26 | 43.84 | 23557| 4679|
>
> The Codestral Mamba-7B model did not perform well with EMR. It consistently generated unexpected special tokens (e.g., 'Станов'), which did not appear in our training data and significantly impacted performance. We think this issue arises because Codestral Mamba-7B is inadequately pre-trained and struggles with generating relatively long texts required for our contextual summaries.
>
> For RetNet, the pretrained models we accessed were trained on a limited number of tokens (several billion), which is also insufficient for conducting post-training with EMR inference.
>
> Additionally, both Mamba(2) and RetNet are variants of linear attention models, with RWKV-7 [7] being the latest state-of-the-art linear attention model. RWKV-7 outperforms both Mamba and RetNet on various benchmarks, utilizing advanced features such as the delta-update rule. As such, we believe RWKV-7 provides a strong representation of the current state-of-the-art recurrent/SSM baselines, which effectively demonstrates the power of our approach.
>
> We anticipate that future releases of well-pretrained Mamba and RetNet models will offer valuable comparison points, and we plan to include them in future work.
>
> comment: We measure the throughput by maximizing the batch size that fits in 80GB GPU memory for both models. Note the throughput is just for reference, as mamba is not highly optimized as swa model. Furthermore, as inference engine and kernel optimizations are time-consuming and out of the scope of this paper, we leave them for future work.
>
> [1] RWKV-7 "Goose" with Expressive Dynamic State Evolution, 2025
>
> ### Q2: Seed & Confidence Interval Results
>
> Thank you for the suggestion. We will rerun the experiments using five seeds and include 95% confidence intervals (CIs) in our final submission. Note that, due to changes in our institution's cluster platform and GPU types, the results in this rebuttal may differ slightly from those in the original paper. However, the overall trends and conclusions remain unchanged.
>
> | Model| LongBench    | NIAH(avg of 8k,16k,32k) | NIAH(avg of 64k,128k,256k) |
> | - | - | - | - |
> | Qwen-2.5-3B-OR  | 46.79 ± 0.01 | 98.20 ± 0.00| 30.85 ± 0.04|
> | Qwen-2.5-3B-NMR | 48.14 ± 0.20 | 86.52 ± 0.12| 68.36 ± 1.50|
> | RWKV-7-3B-OR    | 41.05 ± 0.05 | 92.01 ± 0.14| 12.27 ± 0.10|
> | SWA-3B-4k-OR    | 41.34 ± 0.02 | 29.25 ± 0.15| 3.32 ± 0.21|
> | RWKV-7-3B-EMR   | 47.31 ± 0.11 | 95.88 ± 0.49| 51.87 ± 0.31|
> | SWA-3B-4k-EMR   | 49.47 ± 0.11 | 99.77 ± 0.05| 98.60 ± 0.10|
>
> The results show strong stability with very small confidence intervals (CIs). This stability is primarily due to our use of greedy sampling for evaluation and top-k sampling is only tried when generating over-long responses (the number of samples    is very small). As a result, sampling-related randomness has minimal impact. We will revise our paper to include these CIs in the final version.
>
> ### Q3: Clearer Summary of EMR Details
>
> Thank you for your suggestion to clarify the details of EMR. We will include a more explicit summary of the EMR method in the main text. Due to layout considerations, this update will be made in the final version of the paper.

---

> ### Author Response · Authors · 2025-11-22
> **Response to Reviewer aS6X (2/2)**
>
> ### Q4: Wall-clock Time and FLOPS Decomposition
>
> As mentioned, due to changes in our institution’s cluster platform and GPU types, we re-evaluated the wall-clock time of Qwen-3B-OR and SWA-3B-4k-EMR on the new platform using 500 samples with a 64k context length. The results are summarized below.
>
> | Model| Time (old cluster) | Time (new cluster) |
> | - | - | - |
> | Qwen-3B-OR    | 1500 (1x)| 1032 (1x)|
> | SWA-3B-4k-EMR | 797 (0.53x)| 593 (0.57x)|
>
> We then analyzed the time and FLOPS breakdown during the prefilling and decoding stages.
>
> | Model| Prefilling Tokens | Prefill Throughput (token/s) | Prefill Time (s) | Decoding Tokens | Decode Throughput | Decode Time (s) | Total Time (s) | FLOPS (T)   |
> | - | - |- | - | - | - | - | - | - |
> | Qwen-3B-OR    | 65653| 27273 (1x)| 2.41| 32| 494 (1x)| 0.06| 2.47 (1x)| 483 (1x)|
> | SWA-3B-4k-EMR | 65653| 59283 (2.17x)| 1.11| 6381| 12829 (26x)| 0.50| 1.61 (0.65x)| 204 (0.41x) |
>
> The major time-saving occurs during the prefilling stage, where SWA-3B-4k-EMR achieves 2.17x throughput over Qwen-3B-OR, saving 1.3 seconds per sample. However, the decoding stage introduces overhead, with SWA-3B-4k-EMR requiring an additional 0.44 seconds compared to Qwen-3B-OR, due to processing a significantly larger number of decoding tokens (6381 vs. 32). That said, SWA-3B-4k-EMR demonstrates a 26x higher decoding throughput, making EMR efficient in long-context generation. This efficiency is largely due to the reduced FLOP cost of SWA and its ability to use larger batch sizes during decoding, thanks to its low memory consumption.
>
> In total, SWA-3B-4k-EMR achieves 0.65x the total time compared to Qwen-3B-OR per sample, though in practice it achieves 0.57x with 500 samples. This discrepancy likely results from additional overhead caused by Qwen-3B-OR's higher memory usage and inference engine operations, which are not accounted for in the decomposition. This highlights the greater practical speedup potential of SWA-3B-4k-EMR.
>
> Furthermore, the FLOPS of SWA-3B-4k-EMR is reduced to 0.41x compared to Qwen-3B-OR, indicating substantial room for further optimizations in implementation. As kernel and inference engine optimizations are time-consuming and beyond the scope of this paper, we leave them for future work.
>
> ### Q5: What default chunk:window ratios do you recommend, and how sensitive is accuracy/speed to this ratio across tasks?
>
> Based on our experiments, we recommend a chunk:window ratio of approximately 1:2. This ratio strikes a good balance between accuracy and speed, making it a solid starting point for hyperparameter tuning.
>
> We evaluated SWA-3B-4k-EMR using three different chunk:window ratios (1/4, 1/2, and 3/4) on Longbench. We found that four tasks were significantly impacted by the ratio, with differences exceeding 5 points. For the other tasks, the mean of maximum difference of tasks was just 1.58 points. The tasks most affected typically require maintaining more information, where a smaller chunk size helps.
>
> | Chunk:Window Ratio| 2wikimqa | multi_news | passage_retrieval_en | passage_count |
> | - | - | - | - | - |
> | 1:4| 50.84| 22.83| 99.00| 32.00|
> | 1:2| 52.97| 13.72| 96.00| 28.00|
> | 3:4| 46.19| 6.40| 91.50| 21.50|
>
> Regarding speed, all tasks show sensitivity to the chunk:window ratio, with the time consumption increasing as the chunk:window ratio decrease. This finding is consistent with the results in Table 4 of the paper, where a larger ratio leads to more efficient inference.
>
> ### Further Discussion on Co-optimization
>
> We appreciate your attention to our method's performance on Mamba and the inference time decomposition. We agree that further explanation of why **Co-optimization** is so critical for recurrent LLMs is necessary.
>
> As highlighted, SWA is much more efficient than SA during the decoding stage (26x) compared to the prefilling stage (2.17x). This suggests that long-generation is a crucial optimization direction for improving recurrent LLMs, as we have shown in our work. Additionally, long-generation requires the model to possess a strong generation capability, which underscores the importance of well-pretrained base models—evidenced by the failure of Mamba-EMR in this regard.
>
> Thus, we view the failure of Mamba-EMR not as a limitation of our EMR method but as a sign of the importance of **co-optimization**. Previous research has mainly focused on "light pretraining + simple one-step inference evaluation," which may not fully capture the capabilities of recurrent LLMs. Our paper represents a significant step forward in jointly considering architecture and inference co-optimization for recurrent LLMs. We believe this approach can be further expanded, with improvements in base model pretraining, inference strategies, training data, and more.
>
> ---
>
> Thank you once again for your constructive feedback. We hope these updates and clarifications address your concerns. We would be grateful if you would consider revising your score based on the additional information provided.

---

### Author Response · Authors · 2025-12-01
**Rebuttal Summary**

Dear Area Chair,

Thank you for your time and effort reviewing our submission. Because the discussion was cut short by the information leak, we summarize here the key points of our rebuttal and the additional material we provided.

## Our main contributions
- Practical impact: Long-context understanding is a critical bottleneck for recurrent LLMs. We propose a novel method that substantially narrows the gap: recurrent LLMs can achieve comparable long-context understanding performance to self-attention LLMs. We believe this result can increase confidence in recurrent LLMs and attract investment to make them more practical and widely adopted.
- Methodology: We introduce an architecture–inference co-design for recurrent LLMs that departs from prior work focused primarily on architecture changes. Our approach reframes the problem from “light pretraining + one-round inference” to “stong pretraining/post-training + agentic multi-round inference.” We argue this is a paradigm shift that better unlocks the potential of recurrent models.

## Responses to reviewers aS6X, DGWQ, and 8gR7
Reviewers aS6X, DGWQ, and 8gR7 acknowledged the significance and novelty of our method and raised technical questions. In response we provided:
- Results across different random seeds and confidence intervals,
- Decomposition of wall-clock time and FLOPs to support our inference-efficiency claims,
- Sensitivity analysis on the chunk:window ratio,
- HELMET benchmark results,
- Comparisons with Mamba and so on.

The only missing experiment is that applying end-to-end multi-round inference to Mamba and RetNet, which is raised by Reviewer aS6X. We explained that multi-round inference requires well pre-trained base models to be effective. We were unable to obtain adequately pre-trained Mamba/RetNet models because they currently generate poor long texts. This limitation underscores a broader issue: recurrent LLM research is in a feedback loop: weak long-context performance -> limited investment and therefore insufficient pretraining, which in turn prevents effective post-training and multi-round inference -> weak long-context performance than self-attention LLMs.
Our work breaks this deadlock by demonstrating that with appropriate co-optimization, recurrent LLMs can match self-attention LLMs on long-context understanding—thereby providing a strong incentive for further pretraining and adoption of agnetic multi-round inference. We believe this new feedback loop will significantly accelerate the development of recurrent LLMs.

It's a pity that we could not receive feedback from these three reviewers. However, we believe our responses and additional experiments adequately address their concerns.

## Responses to reviewers AYW3

Reviewer AYW3 raised two concerns:
- Why not provide multi-turn conversational benchmarks: The reviewer’s concern stems from a misunderstanding of our goal. Our focus is long-context understanding rather than multi-turn conversational modeling; the reviewer agreed after clarification.
- What's our novelty vs. KV-cache compression methods: We provided detailed clarifications on methodology, computational complexity, infrastructure compatibility, and included pseudocode to illustrate concrete differences. We do not receive final feedback from the reviewer about this point, but we think this explanation clearly demonstrates the novelty of our approach. Besides, other reviewers concurred that our method is novel.

In conclusion, we are grateful to the reviewers and the area chair for their constructive feedback and efforts, especially given the unexpected interruption of the discussion phase. Then, we believe the remaining technical issues do not undermine the novelty or significance of our contribution, and that our responses address the reviewers’ concerns. We respectfully ask you to consider this work for acceptance.

Sincerely,
The authors

---

### Meta-Review · Area_Chair_SNtW · 2026-01-04

**Summary:**

The paper presents a method for improving long-context understanding, by proposing a new inference procedure, named “Smooth Reading” (also referred as End-toEnd Multi-Round inference (EMR). Instead of going through the whole context once, the model now reads input in chunks. After reading each chunk, it emits its contextual summary, and updates hidden memory. They focus on applying their method to Recurrent LLM, and their experiment shows that their approach close the performance gap between Recurrent and Self-Attention LLM (evaluated on LongBench and NIAH task).

The reviewers had some concerns about experimental procedures (e.g., variance with different random seeds, wall-clock time comparison, sensitivity to hyper parameters), and authors have mostly provided satisfying answers. They also provided new results on another dataset (HELMET), which shows promising results. Reviewer had some questions about comparison with other approaches (mainly, KV compression line of work), which authors have provided good responses. While additional results on applications to other architecture (e.g., Mamba), as suggested by reviewers, would strengthen the paper, but I do not view it as a necessary component. The method is clearly described, new, and shows convincing experimental results.


Minor:    The description of training data is not very easy to identify. I’d recommend authors to update the draft with more clear description. Revier aS6X also mentions "EMR details are spread across §3 and A.1".

**Reviewer Concerns:**

See the meta review.

**Reviewer Scores:**

The authors provided many additional results answering questions from reviewer aS6X, as well as for DGWQ, so I think they would have increased the score. I do not think the other two reviewers would have changed the scores.

---

### Decision · Program_Chairs · 2026-01-26

Accept (Poster)